# Identification and Preliminary Hierarchisation of Selected Risk Factors for Carbapenemase-Producing Enterobacteriaceae (CPE) Colonisation: A Prospective Study

**DOI:** 10.3390/ijerph20031960

**Published:** 2023-01-20

**Authors:** Małgorzata Timler, Wojciech Timler, Ariadna Bednarz, Łukasz Zakonnik, Remigiusz Kozłowski, Dariusz Timler, Michał Marczak

**Affiliations:** 1Department of Management and Logistics in Healthcare, Medical University of Lodz, 90-419 Lodz, Poland; 2Faculty of Economics and Sociology, University of Lodz, 90-214 Lodz, Poland; 3Department of Emergency Medicine and Disaster Medicine, Medical University of Lodz, 92-212 Lodz, Poland

**Keywords:** multi-drug resistance, asymptomatic carriers, culturing, risk factors, carbapenemase-producing Enterobacteriaceae

## Abstract

Drug-resistant bacteria are one of the main reasons of deaths worldwide. One of the significant groups of these bacteria are carbapenemase-producing Enterobacteriaceae (CPE). The goal of this cross-sectional study was the identification and hierarchisation of selected risk factors of CPE colonisation. To achieve that goal, we examined 236 patients for the presence of CPE using the standard method of anal swabs. The patients were divided into three groups: hospitalised patients; those chronically dialysed; those requiring home care. A very thorough medical interview was conducted for comorbidities. A statistical analysis relationship between comorbidities and locations of the patient’s stay with the positive result of the culture was investigated. A significant relationship was demonstrated between the positive result of the culture and confirmed dementia, heart failure, connective tissue diseases, and established irregularities in the level of leukocytes. No significant relationship was demonstrated with the remaining comorbidities considered in the study. Afterwards these factors were compared for importance for the assessment of risk of a positive swab result—the biggest importance was found in establishing connective tissue disease. Next were dementia, abnormal values of leukocytes, heart failure, and at the end, stay at the orthopaedics ward. Conclusions: The study identified asymptomatic carriers of CPE, which demonstrates the need for further studies in order to identify infection risk factors. The connective tissue diseases are the most important variable which enable the prediction of CPE colonisation—the next ones are dementia, abnormal values of leukocytes, heart failure, and stay at the orthopaedics ward.

## 1. Introduction

### 1.1. Background

The growing bacterial antimicrobial resistance (bacterial AMR) is a real hazard to human life and health globally.

Global burden of disease (GBD) is the notion of the burden caused by diseases worldwide. It concerns the assessment of global population’s health in the analysis of the number of cases of disability and number of deaths for various causes worldwide. In February of 2022 the international group of scientists working on GBD performed a systematic overview of the scientific literature, which indicated that AMR is the third cause of deaths worldwide, after coronary artery disease and stroke [1].

The development of antibiotic resistance by strains of pathogenic bacteria results in the restriction of the possibility of treatment, and as a consequence leads to increased mortality. This problem is becoming one of the largest public health challenges. This phenomenon applies worldwide; however, it differs in the frequency of occurrence between continents and countries [2,3,4,5].

### 1.2. Current State of the Subject

On 26 January 2022, a first report by the ECDC and WHO (World Health Organisation) was published on the problem of drug resistance of microbes in Europe. The report states that every year in the European Union (EU/EEA) countries there are 670,000 infections caused by antibiotic resistant bacteria, and 33,000 people die as a result thereof. The health care costs have been established to amount to 1.1 billion Euro [6].

Drug resistance may apply to multiple groups of drugs (MDR, multi-drug resistance) [7]. The definition of MDR for Gram-positive and Gram-negative bacteria means resistance to three or more antibiotics. Drug resistance may apply to all available drugs (PDR, pan-drug resistance). The phenomenon of extended drug resistance (XDR) is also known, which means susceptibility to no more than two groups of drugs [8,9]. Multi-drug resistant organisms (MDRO) are also called alert pathogens. [9].

The name Enterobacteriaceae applies to one of the seven families of the Enterobacterales order, the main representatives of which are Escherichia, Salmonella, Shigella, Enterobacter, Klebsiella, Citrobacter [10]. It is a family of clinically significant Gram-negative bacteria widely spread all over the globe. They are present in water and soil, are an element of proper gastrointestinal flora of animals and humans, and infection may occur via droplets, food, and by contact with a contaminated surface. In some circumstances the Enterobacteriaceae may cause serious infections, in the treatment of which carbapenems were effective until recently.

The overuse and unjustified use of carbapenems in the treatment of severe infections caused by Gram-negative bacteria from the Enterobacteriaceae family has led to the development of carbapenem resistant Enterobacteriaceae (CRE). The ability to produce carbapemenase, is a trait of carbapenemase producing Enterobacteriaceae (CPE). Carbapenemases are enzymes able to hydrolyse penicillins, cephalosporins, and carbapenems, antibiotics which are called “drugs of last resort”.

CRE infections are diagnosed in health care facilities, mainly in in-patient treatment facilities. This does not mean that all cases of CRE are detected, and one of the reasons is that the presence of carbapenemase producing Enterobacteriaceae is not always related to the presence of clinical symptoms. This condition is called colonisation/being an asymptomatic carrier [11]. In NPOA it was assumed that the duration of CPE colonisation observed from the time when CPE was detected amounts to 6 months. According to some sources the duration of colonisation is in the range of 43 to even up to 387 days [12].

Based on the analyses the factors predicting CPE colonisation include the following:extended hospitalisation, in some of the literature the length of hospitalisation exceeding 20 days appears [2], according to other data it is 14 days [13],hospital stay during the last 12 months,treatment during the last 3 months in health care facilities in countries with endemic and unknown frequency of occurrence of CPE,antibiotic treatment during the last 10 days [2,14].

### 1.3. What Remains to Be Known about the Subject

Despite the high number of deaths and the long-time of cpe colonisation of asymptomatic carriers, there is still a lack of research on colonisation, infections, and communication between health care facilities in the category of notification about asymptomatic carrier status.

## 2. Materials and Methods

### 2.1. Research Tasks

The goal of the study was to identify risk factors for CPE colonisation in selected groups of patients. In order to implement this goal, the following research tasks were formulated:Dividing patients into groups according to accepted criteriaEstablishing the minimum size of the studied groupIdentifying comorbidities in patients from the studied groupsChecking the conformity of the distributions of analysed variables with normal distributionChecking the relationship between the distinguished factors and a positive result of a swab for CPEEstablishing the impact of individual factors on the swab’s result

### 2.2. Study Design

The cross-sectional study examined 236 patients in the period from 1 January 2020 until 31 December 2020 at the following health care facilities:❖GP care centre Poradnia Lekarzy Rodzinnych MEJAmed Sp. z o.o. Łódź, ul. Przybyszewskiego 32/34 in Łódź❖M. Kopernik Province Multi-Specialist Oncology and Traumatology Centre in Łódź, 93-513 Łódź, ul. Pabianicka 62❖WAM University Clinical Hospital—Central Veteran’s Hospital, 90-549 Łódź, ul. Żeromskiego 113

### 2.3. Selection, Sample and Procedure

Multiple groups were formed from among the patients participating in the study:The first group were hospitalised patients, who were not tested for carbapenemase producing Enterobacteriaceae (CPE) as a result of an infection.The second group included patients treated under home conditions and requiring care.The third group included chronically dialysed patients.

The patients, after providing written consent, were tested for the presence of carbapenemase producing Enterobacteriaceae (CPE) by a standard method anal swab. The material collected from the patients was transported to the Synevo Microbiological Laboratory in Łódź, at ul. Krakusa 28, which was selected for participation in this study in a tender procedure. The swabs arrived at the laboratory on a typical transport medium in the form of coded (anonymised) samples, without access to personal data of the patients.

In the laboratory the procedure of handling the provided biological material was as follows:Inoculation of material (anal swab) onto chromogenic media (Agar ChromID CARBA, BioMerieux).Incubation at a temperature of 37 °C for 24 h.For all growing colonies identification with the MALDI TOF method (Vitek MS, Biomerieux).For Gram (-) rods of the Enterobacterales order:
antimicrobial susceptibility testing for carbapenems (ertapenem, meropenem) using the disc diffusion test.conducted phenotypic testing for detection of carbapamenases in accordance with the recommendations of the National Reference Centre for Antimicrobial Susceptibility Testing of Microorganisms (KORLD, Krajowy Ośrodek Referencyjny ds. Oznaczania Lekowrażliwości Drobnoustrojów) (“Detection of carbapanemases—recommendations 2017”), available at the website www.kordl.edu.pl).In case a positive result was obtained (carbapamenases were detected) the strain was sent to KORLD to obtain confirmation.

### 2.4. Analysis

In order to achieve results that could be generalised, and wider conclusions could be formulated the minimum sample size was determined [15]. According to research by GUS, during the study a total of 38,382,600 persons were living in Poland [16]. When taking into account the size of the sample, a value for a normal distribution of 1.96 was calculated (assuming a statistical significance of α = 0.05). As the assumed maximum error value, a range of 5–10% was accepted. As a final result it was established that for error d = 5% the sample size should amount to 384 persons. For an error with a size of d = 10% the sample size should amount to 96 persons. Therefore, the study assumed that an attempt would be made to collect swabs from approximately 300 persons. Finally, as already mentioned, a total of 236 results were obtained. The sample size significantly exceeded the minimum threshold value calculated using the formula (which was in the range of 96–384).

The analyses of the collected material were conducted with the IBM SPSS Statistics v.27 software (academic license, Armonk, NY, USA).

## 3. Results

Due to the difficulty in conducting the study, resulting from the fact that it was performed during the COVID-19 pandemic, the data were collected primarily from the patients of four hospital wards and from persons dialysed at the hospital outside of the listed wards (data contained in Table 1). There was an attempt to keep an even size of the groups. Unfortunately, for three places of stay we managed to collect a lower number of results than assumed (Haematology, Orthopaedics, and dialysed outside of the wards).

To maintain the most representative character of the obtained results, the patients were randomly subjected to the study. After data collection, it turned out that the sex structure of the study subjects was fairly uniform (Table 2), although generally a slight predominance of men were observed. As could have been expected, in the 45–54 and 55–64 groups a clear predominance of men was visible, which then changed to a predominance of women (65+ group)—this condition could be justified by the longer life expectancy of women over men in Poland. The comparison of the general age structure obtained in the study with the age structure of persons hospitalised in Poland in a comparable period (year 2020) is an argument for the representativeness of the study. Differences between individual groups do not exceed five percentage points. Taking into account the observations described below and the significant troublesome problems in the performance of the study, resulting from the COVID-19 pandemic, and also considering the number of the studied cases, it may be assumed that the results of the study should form the basis for formulating valuable final conclusions.

The study—in relation to the planned future analyses—collected information about the health status of patients. This information frequently constitutes independent variables, the impact of which on the ability to obtain a positive swab result (dependent variable) was verified. The most frequent comorbidity of the patients (Table 3) was arterial hypertension, observed in more than half of the study subjects (54.8% of the study subjects), the next one was chronic kidney failure (in 36.7%). Almost every fourth study subject had type 2 diabetes (23.5%), and every fifth suffered from heart failure (20.8%). Insulin therapy was used in 10.4% of the studied patients. Thyroid diseases and tumours occurred in patients less frequently (17.6% each), and leukaemia, of which 13.1% of the study subjects suffered from, even less frequently. A number of 6.8% of patients had suffered a stroke, peptic ulcer disease, and peripheral vascular disease, and 5.4% of patients had COPD and liver disease. Other diseases were observed more rarely—dementia in 4.1%, connective tissue diseases in 3.2%, hemiplegia in 2.7% of study subjects. Four persons (1.8% of patients participating in the study) were treated for type 1 diabetes, and three persons (1.3%) for lymphoma. Tumour metastases were observed in 3.2% of patients. Two persons (0.9%) had had a kidney transplant.

In order to process the data descriptive methods statistical inference methods were used. To establish a specific method, the conformity of the distributions of analysed measurable variables with the standard deviation was checked. This was performed using the Shapiro–Wilk test [18]. Because the distributions of analysed measured variables differed significantly from a normal distribution, nonparametric tests were used instead of parametric tests. To compare two groups of patients (e.g., with a positive and negative result of an Enterobacteriaceae test) the Mann–Whitney U test was used [18].

The Mann–Whitney U test is the strongest nonparametric alternative to the Student’s t-test for independent samples. The results with an appropriate number of degrees of freedom and probability of error *p* < 0.05 were considered statistically significant. The Mann–Whitney U test was used for all possible variables obtained in the study. This was performed using the IBM SPSS Statistics v.27 software (selecting Analysis > Nonparametric tests > Independent samples; with the option of the Mann–Whitney U Test).

The pair of hypotheses were defined as follows:

**H0:** 
*The distribution of the studied factor is the same for the category positive swab = 1 no = 0.*


**H1:** *~H0*.

The obtained results are presented in Table 4 below:

As one can see pursuant to Table 4, in most cases no association between a positive swab result and the presence of the given factor can be demonstrated. However, this situation occurs in four cases: in the case of confirmed dementia, heart failure, connective tissue diseases, and established abnormalities in the level of leukocytes (result out of normal range). Each established association was marked with a grey background in the table. In each of the mentioned cases significance did not exceed the assumed level of 0.005. However, sometimes in the literature it has been suggested that in addition to the statistical significance measured by the value of 0.05 one can speak of some statistical tendency [19,20]. This tendency is present when the significance level exceeds 0.05 but is not larger than 0.1. In the case of the used test this situation occurred when the study was conducted at the orthopaedics ward (marked with a less intense colour of the row background in the analysed table). The Mann–Whitney test also enables the relative importance of individual variables to be established in the shaping of the risk of a positive swab result [18]. The procedure which enables this is selected in the IBM SPSS Statistics software (Armonk, NY, USA) through Analysis > Nonparametric tests > Traditional tests > Two independent samples and indicating the Mann–Whitney test. As a result, we obtain Table 5, which includes, among other data, the value of the Z-score. The values of this score are compared between variables, which enables creation of a ranking of traits due to their influence on the swab result variable.

The order of significance of risk factors was established based on quantitative analysis. Further order was of the experts’ choice and initial rankings.

As one can see in the table above, the highest value—and therefore the relatively highest importance for the risk of a positive swab result—is the diagnosis of a connective tissue disease (a Z-score of 3.593). The next factors in sequence are as follows: dementia, abnormal values of leukocytes, heart failure, and at the end, stay at the orthopaedics ward.

In global studies an alternative is proposed, therefore, in order to additionally ensure the credibility of the results Kendall’s tau-B correlation coefficient statistical analysis was used.

As one can see, the results concerning significance level in Table 6 are in line with the information provided in Table 4. Of course, the relationships were established for exactly the same variables.

## 4. Discussion

The fight against the spread of CPE is being conducted on many levels. Research is introducing new diagnostic methods and treatments. There are known methods for limiting the spread of this bacteria in a situation of a confirmed infection. The largest group among the studied patients were hospitalised patients. It is in hospitals where the cases of CPE are most frequently diagnosed, that is, infections with clinical symptoms or asymptomatic, the latter situation being referred to as an asymptomatic carrier. The rate at which CPE bacteria spread between patients present in their close vicinity, called the acquisition rate, is 3.2% [21].

Among the hospitalised patients the ones treated at the haematological, orthopaedics, and nephrology wards and at the Intensive Care Unit (ICU) were selected. All indicated wards have a higher frequency of CPE diagnosis.

The patients of haematological wards are a known CRE risk group. Blood borne infections caused by CRE are the leading cause of morbidity and mortality among haematological patients. Based on active screening tests for CRE conducted in China among haematological ward patients the frequency of CRE colonisation was established at a level of 16.46%.

Intensive Care Units are included among wards with a high risk of CRE colonisation and infection [2,22,23,24]. Patients arriving at these wards are in severe conditions, suffering from multiple diseases, and are subjected to intensive treatment with, among others, antibiotics, mechanical ventilation, catheters, and other forms. Mechanical ventilation is a risk factor for CRE colonisation [25]. All study participants were burdened with comorbidities. Many researchers use various indicators when assessing this risk factor. In Michigan, in a study conducted between 1 May 2017 and 13 December 2018 in long-term care facilities a total of 18,216 swabs were collected from patients obtaining a positive KPC result in 2643 cases. In 4.3% of tested patients with a positive result infection developed. In a conducted analysis it was observed that the risk of the infection developing is increased in patients with comorbidities and, in the case of previously diagnosed depression, with low level of albumins [26].

No data were obtained indicating the risk of CPE colonisation in patients burdened with most of these morbidities. In the study a positive result for CPE was significantly more frequently observed in the group of patients with dementia. What is interesting, despite the fact that patients from the outpatient clinic were significantly older, when compared to participants in the remaining groups and that they were more frequently diagnosed with dementia, no CPE colonisation was noted in this group. It may be thus inferred that dementia itself is not a predisposing factor for CPE colonisation, there must therefore exist additional circumstances, such as hospitalisation and its duration. Most frequently a patient in whom CRE is found has a history of long hospitalisation [12], the reason for which is due to diseases unrelated to CRE. The treatment of these diseases entails the use of various antibiotics [12], as a result leading to resistant strains of CRE. However, until now it has not been possible to establish a limit of the number of days of hospital stay above which the risk of occurrence of CPE cases increases. Some of the sources indicate >20 days of hospitalisation [2] as a risk factor, some >14 days [13]. In the case of low-income and middle-income countries (LMICs) the carbapenem resistance appeared already at a hospitalisation duration of 3–6 days [25].

In this study, systemic diseases had a significant importance as CPE carrier risk factor. These are severe diseases with an autoimmune background, leading to many disorders, and their treatment may cause unanticipated complications.

Reduced patient immunity is included among CRE risk factors [25].

Based on research by the Consortium on Resistance Against Carbapenem in Klebsiella and other Enterobacteriaceae (CRACKLE) conducted in Ohio, the CRE colonisation risk factors include the following: female sex, advanced age, previous treatment with antibiotics, stay in healthcare facilities, additional diseases, travelling to endemic regions, presence of invasive devices and drains [26,27,28]. In studies conducted in patients with a Klebsiella pneumoniae infection it was demonstrated that male sex, advanced age, dialyses, post-transplantation state, chronic liver disease, and cancer are factors which predispose for the development of an infection [29]. Additional CRE infection risk factors include the presence of implants and implantable devices in the patient, dialysis, health condition with comorbidities with significant role of diabetes, previous endoscopic examination [2,22,24,30,31].

In addition to polymorbidity (in a wide sense), predictive factors for being a CPE carrier may include drugs and other phenomena. There are studies which indicate that a change of the gut microflora by previous exposition to proton pump inhibitors (PPI) tripled the risk of CPE colonisation in patients from acute-care hospitals (ACH) [32]. In France, after an analysis of material covering the years 2010–2016 an increase in the number of CPE cases by 30% in the autumn was observed, which may demonstrate the seasonality of CPE incidence [33].

Based on multiple analyses it can be seen that the factors predisposing for the development of infection and for colonisation are very similar. Most studies on CRE have concerned hospital environment, and currently we have confirmation of CPE presence in long-term care facilities (LTCFs). There is no data on CRE among persons without contact with health care facilities [11]. Lack of data does not mean that asymptomatic CPE carriers are not present outside of the hospital system; this is indicated by analyses in facilities where screening tests for CPE were introduced for all persons admitted to the hospital.

The conducted analysis, just like many other studies performed so far, does not allow the creation of a CPE colonisation risk factor template, on the basis of which the probability of being a CPE carrier could be calculated. For the moment we rather have a group/set of factors which may increase the risk of being a CPE carrier. This study has many limitations, some of which were caused by the pandemic. Initially a participation of a much larger group of study subjects was assumed, unfortunately the epidemic situation and the patients’ fear of contact with health care facilities reduced the participation of patients. Changes to the structure of health care facilities also had an impact on the groups of patients participating in the study. During the pandemic the awareness of medical personnel of hand hygiene, of the use personal protective equipment, and of following the sanitary and epidemiological recommendations increased significantly, and it is difficult to estimate whether it had any impact on the outcome of the study.

In the fight against the spread of CPE, in addition to further work on establishing a risk factor template it seems justified to emphasise screening tests for CPE. There is an insufficient number of studies on CPE colonisation in healthy persons having no contact with health care systems. The results confirm that early detection, and then appropriate supervision decrease the number of hospital drug-resistant infections [34].

## 5. Conclusions

As a result of the conducted analysis the following conclusions were formulated:A strong correlation of CPE asymptomatic carrier status was observed only with dementia, heart failure, connective tissue diseases, and having a leukocyte level deviating from the norm. A weak correlation of CPE asymptomatic carrier status was observed with a stay at the orthopaedics ward.In the studied group of patients CPE asymptomatic carriers were identified, which indicates the need for further research to better identify the risk groups for a positive result of a test for CPE.

## Figures and Tables

**Table 1 ijerph-20-01960-t001:** The swabbed patient’s place of stay.

The Patient’S Place of Stay	Number	%
Hospital—ICU	67	28.39%
Hospital—Haematology	30	12.71%
Hospital—Orthopaedics	34	14.41%
Hospital—Nephrology	61	25.85%
Hospital—Another ward	1	0.42%
Dialysed in hospital as outpatient procedure	29	12.29%
Outpatient clinic	14	5.93%
Total	236	100.00%

Source: Own work.

**Table 2 ijerph-20-01960-t002:** Age and sex characteristics of the studied group.

Age Category [Years]	Women [n]	Men [n]	Number of Patients [n]	% of the Study	% of Hospitalised Generally in Poland ^a^
65+	66	53	119 ^b^	50.85	45.87
55–64	19	28	47	20.09	17.68
45–54	9	15	24	10.26	11.31
35–44	7	10	17	7.26	11.26
20–34 ^b^	14	13	27	11.54	13.88
Total	115	119	234 ^b^	100 ^bc^	100 ^c^

^a^ Estimated age structure for persons hospitalised in Poland in 2020 [17]. ^b^ In the study one person (women) of an age below 20 years appeared. She was not included in the table due to the fact that the estimations of the age structure of the persons hospitalised in Poland included a 15–19 age category (and this group was not included in the study). One person from the 65+ group was also not included, since the information about their sex was accidentally omitted during the collection of data. ^c^ The added-up data do not include the age group below 20 years of age (so, generally, minors). Source: Own work.

**Table 3 ijerph-20-01960-t003:** Comorbidities of the patients in the studied group.

Comorbidities	Number of Study Subjects [N]	Percentage of the Study Subjects [%]
Thyroid disease	39	17.6
Arterial hypertension (HA)	121	54.8
Type 1 diabetes	4	1.8
Type 2 diabetes	52	23.5
Insulin therapy	23	10.4
TIA in medical history	-	-
Dementia	9	4.1
AIDS	-	-
COPD	12	5.4
cerebral stroke	15	6.8
Heart failure	46	20.8
Peripheral vascular diseases	15	6.8
Connective tissue diseases	7	3.2
Peptic ulcer disease	15	6.8
Liver diseases	12	5.4
Hemiplegia	6	2.7
Chronic kidney failure	81	36.7
After a kidney transplant	2	0.9
Tumour	39	17.6
Metastases	7	3.2
Leukaemia	29	13.1
Lymphoma	3	1.4

Abbreviations HA—Arterial hypertension, TIA—Transient Ischaemic Attack, AIDS—Acquired Immunodeficiency Syndrome, COPD—Chronic Obstructive Pulmonary Disease. Source: Own work.

**Table 4 ijerph-20-01960-t004:** Relationship between the studied factors and the positive swab category, based on Mann–Whitney U Test.

Null Hypothesis	Significance a,b	Decision
The distribution ICU = 1 no = 0 is the same for category positive swab = 1 no = 0.	0.312	Assume H0
The distribution nephrology = 1 no = 0 is the same for category positive swab = 1 no = 0.	0.450	Assume H0
The distribution orthopaedics = 1 no = 0 is the same for category positive swab = 1 no = 0.	0.060	Assume H0
The distribution haematology = 1 no = 0 is the same for category positive swab = 1 no = 0.	0.982	Assume H0
The distribution outpatient clinic = 1 no = 0 is the same for category positive swab = 1 no = 0.	0.471	Assume H0
The distribution of age (years) is the same for category positive swab = 1 no = 0.	0.257	Assume H0
The distribution of sex (f = 1 m = 0) is the same for category positive swab = 1 no = 0.	0.498	Assume H0
The distribution of number of days in the hospital is the same for category positive swab = 1 no = 0.	0.897	Assume H0
The distribution dialysed = 1 no = 0 is the same for category positive swab = 1 no = 0.	0.642	Assume H0
The distribution surgery in the last year = 1 no = 0 is the same for category positive swab = 1 no = 0.	0.644	Assume H0
The distribution thyroid disease = 1 no = 0 is the same for category positive swab = 1 no = 0.	0.132	Assume H0
The distribution HA = 1 no = 0 is the same for category positive swab = 1 no = 0.	0.773	Assume H0
The distribution of type 1 diabetes is the same for category positive swab = 1 no = 0.	0.697	Assume H0
The distribution of type 2 diabetes is the same for category positive swab = 1 no = 0.	0.915	Assume H0
The distribution insulin = 1 no = 0 is the same for category positive swab = 1 no = 0.	0.328	Assume H0
The distribution dementia = 1 no = 0 is the same for category positive swab = 1 no = 0.	0.002	Reject H0
The distribution COPD = 1 no = 0 is the same for category positive swab = 1 no = 0.	0.492	Assume H0
The distribution stroke = 1 no = 0 is the same for category positive swab = 1 no = 0.	0.511	Assume H0
The distribution heart failure = 1 no = 0 is the same for category positive swab = 1 no = 0.	0.039	Reject H0
The distribution peripheral vascular disease = 1 no = 0 is the same for category positive swab = 1 no = 0.	0.511	Assume H0
The distribution connective tissue diseases = 1 no = 0 is the same for category positive swab = 1 no = 0.	0.000	Reject H0
The distribution peptic ulcer disease = 1 no = 0 is the same for category positive swab = 1 no = 0.	0.511	Assume H0
The distribution liver disease = 1 no = 0 is the same for category positive swab = 1 no = 0.	0.367	Assume H0
The distribution hemiplegia = 1 no = 0 is the same for category positive swab = 1 no = 0.	0.632	Assume H0
The distribution chronic kidney failure = 1 no = 0 is the same for category positive swab = 1 no = 0.	0.493	Assume H0
The distribution after a kidney transplant = 1 no = 0 is the same for category positive swab = 1 no = 0.	0.784	Assume H0
The distribution tumour = 1 no = 0 is the same for category positive swab = 1 no = 0.	0.185	Assume H0
The distribution metastases = 1 no = 0 is the same for category positive swab = 1 no = 0.	0.604	Assume H0
The distribution leukaemia = 1 no = 0 is the same for category positive swab = 1 no = 0.	0.309	Assume H0
The distribution lymphoma = 1 no = 0 is the same for category positive swab = 1 no = 0.	0.735	Assume H0
The distribution of creatinine the same for category positive swab = 1 no = 0.	0.380	Assume H0
The distribution of urea the same for category positive swab = 1 no = 0.	0.950	Assume H0
The distribution of HGB the same for category positive swab = 1 no = 0.	0.822	Assume H0
The distribution of CRP the same for category positive swab = 1 no = 0.	0.964	Assume H0
The distribution of sNIBP the same for category positive swab = 1 no = 0.	0.470	Assume H0
The distribution of dNIBP the same for category positive swab = 1 no = 0.	0.672	Assume H0
The distribution of HA the same for category positive swab = 1 no = 0.	0.649	Assume H0
The distribution pacemaker = 1 no = 0 is the same for category positive swab = 1 no = 0.	0.737	Assume H0
The distribution of normal leukocytes is the same for category positive swab = 1 no = 0.	0.005	Reject H0

a Statistical significance is 0.050; b Asymptotic significance was presented. Source: Own work, based on a table generated by IBM SPSS Statistics v.27 software (IBM, Armonk, NY, USA).

**Table 5 ijerph-20-01960-t005:** Ranking of traits for their influence on the swab result, established with a Mann–Whitney U test.

Tested Value ^a^					
	Orthopaedics = 1 no = 0	Dementia = 1 no = 0	Connective Tissue Diseases = 1 no = 0	Heart Failure = 1 no = 0	Leukocytes Normal
Mann–Whitney U test	691.500	670.000	662.000	594.000	420.000
Z	−1.880	−13.053	−13.593	−12.066	2.810
Asymptotic significance (2-sided)	0.060	0.002	0.000	0.039	0.005

^a^ Grouping variable: outcome result = 1 no = 0; Source: Own work, based on a table generated by IBM SPSS Statistics v.27 software (IBM, Armonk, NY, USA).

**Table 6 ijerph-20-01960-t006:** The strength of relationship between the studied factors and the positive swab category based on a Kendall’s tau-B correlation coefficient.

	Positive Swab = 1 no = 0	
	Correlation Coefficient	Significance (2-Sided)
ICU = 1 no = 0	−0.066	0.312
Nephrology = 1 no = 0	0.049	0.450
Orthopaedics = 1 no = 0	0.123	0.060
Haematology = 1 no = 0	−0.001	0.982
Outpatient clinic = 1 no = 0	−0.047	0.471
Age (years)	−0.061	0.257
Sex (f = 1 m = 0)	−0.044	0.498
number of days at the hospital	0.008	0.897
dialysed = 1 no = 0	−0.031	0.642
surgery within the last year = 1 no = 0	0.031	0.644
thyroid disease = 1 no = 0	0.101	0.132
arterial hypertension = 1 no = 0	−0.020	0.773
type 1 diabetes	−0.026	0.697
type 2 diabetes	0.007	0.915
diabetes treated with insulin = 1 no = 0	−0.066	0.328
dementia = 1 no = 0	0.205	0.002
COPD = 1 no = 0	−0.046	0.492
stroke = 1 no = 0	0.044	0.511
heart failure = 1 no = 0	0.139	0.039
peripheral vascular disease = 1 no = 0	0.044	0.511
connective tissue diseases = 1 no = 0	0.242	0.000
peptic ulcer disease = 1 no = 0	0.044	0.511
liver disease = 1 no = 0	0.061	0.367
hemiplegia = 1 no = 0	−0.032	0.632
chronic kidney failure = 1 no = 0	−0.046	0.493
after a kidney transplant = 1 no = 0	−0.018	0.784
tumour = 1 no = 0	−0.089	0.185
metastases = 1 no = 0	−0.035	0.604
leukaemia = 1 no = 0	0.068	0.309
lymphoma = 1 no = 0	−0.023	0.735
creatinine	−0.053	0.380
urea	−0.003	0.950
HGB	−0.013	0.822
CRP	−0.003	0.964
sNIBP	0.043	0.470
dNIBP	0.026	0.672
HA	0.027	0.649
implanted pacemaker = 1 no = 0	−0.023	0.737
Leukocytes normal	−0.189	0.005

Abbreviations: ICU—Intensive Care Unit, COPD—Chronic Obstructive Pulmonary Disease, HGB—haemoglobin, CRP—C-reactive protein, sNIBP—systolic non-invasive blood pressure, dNIBP—diastolic non-invasive blood pressure, HA—arterial hypertension; Source: Own work, based on a table generated by IBM SPSS Statistics v.27 software (IBM, Armonk, NY, USA).

## Data Availability

The data presented in this study are available on request from the corresponding author.

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
