# Peer review of "Identification and Preliminary Hierarchisation of Selected Risk Factors for Carbapenemase-Producing Enterobacteriaceae (CPE) Colonisation: A Prospective Study"

_ijerph, 2023, doi:10.3390/ijerph20031960_

Round 1

Reviewer 1 Report

This manuscript focused on the risk factors associated with Carbapenemase-producing Enterobacteriaceae (CPE) colonisaiton. Using the standard method of anal swabs, the authors examined 236 patients for the presence of CPE, followed by a thorough medical interview for comorbidities. They linked the positive results to the comorbidities and identified some risk factors, including dementia, heart failure, connective tissue diseases and abnormal leukocyte levels. 

A major concern:

The title mentioned identification and hierarchisation of risk factors, but the manuscript only showed data for the identification of risk factors that show statistical signifance, but nothing about hierarchisation.

A minor concern:

In the Conclusions section, Point #1 summarized the five factors including dementia, heart failure, connective tissue diseases, a stay at the orthopaedics ward, and abnormal leukocyte level. However, among these five factors, staying at the orthopaedics ward is the one that has different significance level compared to the other four, and it may worth listing/describing it separately from them. 

Author Response

Dear Sir/Madam,

The answers to the questions raised beforehand find themselves in the attachment down below.

Best regards,

Authors

Reviewer 2 Report

Thank you very much for allowing me to review your manuscript. The objective of the study is of great relevance, and the subject of great impact, especially if we focus on hospital epidemiology. However, from my point of view, the manuscript in its current version should be improved.

1.-Title: the title should focus on the objective and main result. In addition, the type of study that has been carried out should be stated in the title.

2.- Abstract: it should also report on the type of study that has been carried out. No minimum information is provided to describe the study sample. The main results are not reported.

3.- Keywords: It is striking that "carbapenemase-producing enterobacteriaceae" is not reflected as a keyword. The keywords should be better focused.

4.- Introduction. It should be shortened. It is difficult to read. There are too many paragraphs. For example, it is not necessary to explain what multiresistance is, it is not necessary to explain what Enterobacteriaceae are, etc... Reference can be made to it (it should be) but referring to the bibliography, without going overboard with the explanation. Nor should the secondary objectives be reflected at the end of the introduction (in addition, some of them are not a secondary objective, but part of the method). In short, the introduction should be reduced to 3 paragraphs: 1st paragraph, the background of the topic under study; 2nd paragraph, the current state of the subject, what is known about the subject under study; and the 3rd paragraph, what remains to be known about the subject under study, and how its objective fits into this.

5.- Methodology: It should be better structured, for example in sections such as: 1) study design, 2) Selection, sample and procedure, 3) Ethical considerations, 4) Analysis. It should be better explained how the study patients are accessed. It should be explained if the study has been approved by an ethics committee for research with human beings. Why was a sample size calculated for a descriptive cross-sectional study, if what is intended is to establish risk factors. A sample size should be established for a cohort or case-control study. With a cross-sectional design, as the authors seem to suggest, risk factors could never be estimated. On the other hand, it is not necessary to establish the formula for calculating the sample size, since it is not an academic paper.

6.- Results: The results should be better organized. It is not necessary to explain the formulas used for data analysis. Although the analysis should be better explained in the methodology section. Tables 1, 2 and 3 can be summarized in a single table 1, of characteristics of the study sample. Table 4 should represent the results that explain the main objective, but not reflect the hypotheses of the study. Regarding this table, I am not able to understand why a non-parametric analysis such as the Mann-Whitney U is proposed. From what I can intuit, since the authors do not explain the variables collected in the study, the data is processed as counting variables, and presented as percentages. Therefore, dichotomous variables are continuously being evaluated, so the statistical test to use would be the Chi-square. The same with respect to table 5 and table 6, in which the results of correlation of dichotomous, non-continuous variables are presented. I would be grateful to the authors if they could explain to me how they have proceeded to carry out these analyses.

On the other hand, and leading to the discussion and conclusions, I do not understand how the authors can establish that certain comorbidities or factors are classified as "risk factors". Data is being processed, as far as I could understand, cross-sectional. What the authors define as cause or effect, are collected at the same time. There is no follow-up or longitudinality in the data, therefore, the study lacks a major cause criterion, temporality, so it is very risky from a bivariate analysis, I think wrong, to establish risk factors. In addition, and also as far as I could understand, the hierarchization of the factors classified as risk, is established based on the p-value of Kendall's Tau?

In my opinion, there are multiple points that must be explained and corrected, and that invalidate the results, so the discussion would also be invalid.

Author Response

(The authors gave the same response as above.)

Round 2

Reviewer 2 Report

Dear authors Thank you very much for the answers to my clarifications. I am aware of the good work you have done, and the effort involved in undergoing peer review. Your answers have clarified the doubts that had arisen, however, before any of the changes that you have introduced, I feel obliged to request further clarifications. 1.- You have introduced in the title, as well as in the methodology, that the current work is a longitudinal study. From my point of view, as I mentioned in the first clarifications, it seems to me a study with a cross-sectional methodology. As far as I can understand, you have recruited the patients of the sample during the period of one year (01.01.22 - 12.31.22) in different health care points, and that when the patient agrees to participate in the study, the data is collected needed at the same time. The fact that the recruitment was carried out in a period of one year does not define the study as longitudinal. What defines a study as longitudinal is that the patients are followed up for a period of time, collecting the values of the variables at different moments of the follow-up, and taking into account causality criteria (temporality, directionality and association). Reporting that it is a cross-sectional study (in fact, the sample size was calculated for a cross-sectional study) should be considered, or better information regarding the recruitment and follow-up of patients, since it is not understood all right. 2.- I insist that the 6 methodology points (in the new version, section 1.3) should be removed from the introduction section, and if the authors consider it necessary for this information to appear in the manuscript, do so in the methodology section . 3.- Results: I thank the authors for the clarification in the form of a table on the Chi-square analysis. Thanks. 3.a.- On the other hand, I believe that Table 4 should not report on the action taken on the null hypothesis. I believe that any focus author of this publication understands the decision to be made based on the p-value, which the authors do report. 3.b.- In line 240, the typing error that appears in the name "Manna-Whitney" should be corrected, the last a of "Manna" should be removed. 3.c.- All the analyzes proposed are bivariate analyzes of the relationship between the different variables analyzed. These analyzes make it possible to establish dependency between the variables analyzed, but not to determine the association between them. Based on the analyzes that have been proposed, it could be observed that there is a dependency between presenting a certain characteristic (for example, connective tissue disease) and presenting a positive swab for CPE; or that there is a correlation between these variables. However, these analyzes would not make it possible to determine the association between the variables (understood as a measure of risk). For this, a relative risk measure or Odds Ratio would be necessary. Nor do I understand how you elaborate the ranking, how did you come to the conclusion that connective tissue diseases are the most important risk factor for presenting a positive swab? Do you do it because the p-value of the contrast is closer to 0? If it was done this way, it would not be correct. A simple way to do this would be to assess the risk attributable to each factor and compare them. This would provide an approximation to the impact of each factor on the risk of presenting positive swab. Therefore, conclusions closer to the data of the study and the analyzes carried out should be drawn up. 4.- Conclusions: It should not be concluded that "there is a strong association between CPR asymptomatic carrier status with dementia...", since a true measure of strength of association has not been used. A conclusion closer to the data would be that a correlation or dependency is observed between the variables analyzed.
